# Robustness to Label Noise Depends on the Shape of the Noise Distribution

**Diane Oyen**
Los Alamos National Lab
doyen@lanl.gov

**Michal Kucer**
Los Alamos National Lab

**Nick Hengartner**
Los Alamos National Lab

**Har Simrat Singh**
Los Alamos National Lab

## Abstract

Machine learning classifiers have been demonstrated, both empirically and theoretically, to be robust to label noise under certain conditions — notably the typical assumption is that label noise is independent of the features given the class label. We provide a theoretical framework that generalizes beyond this typical assumption by modeling label noise as a distribution over feature space. We show that both the scale and the *shape* of the noise distribution influence the posterior likelihood; and the shape of the noise distribution has a stronger impact on classification performance if the noise is concentrated in feature space where the decision boundary can be moved. For the special case of uniform label noise (independent of features and the class label), we show that the Bayes optimal classifier for $c$ classes is robust to label noise until the ratio of noisy samples goes above $\frac{c-1}{c}$ (e.g. 90% for 10 classes), which we call the *tipping point*. However, for the special case of class-dependent label noise (independent of features given the class label), the tipping point can be as low as 50%. Most importantly, we show that when the noise distribution targets decision boundaries (label noise is directly dependent on feature space), classification robustness can drop off even at a small scale of noise. Even when evaluating recent label-noise mitigation methods we see reduced accuracy when label noise is dependent on features. These findings explain why machine learning often handles label noise well if the noise distribution is uniform in feature-space; yet it also points to the difficulty of overcoming label noise when it is concentrated in a region of feature space where a decision boundary can move.

## 1   Introduction

An open question in machine learning is how the quality of training data, especially the labels, affects the learned model [2, 4, 13, 22, 24]. As an example shown in Figure 1, we imagine a classification task for detecting cancer based on observed biomarkers. The generative model assumes that there is a hidden variable, called $Y^*$, denoting whether a person actually has cancer or not. Depending on the state of $Y^*$, biomarkers can be observed, denoted as a feature vector $x$, drawn from some distribution $x \sim P[X|Y^*]$. The goal of machine learning (ML) is to predict the presence of cancer given the biomarkers — yet to train and evaluate the ML model, we have diagnoses, $\{y\}$, which may not always be correct ($y \neq y^*$ for some samples). The question of label uncertainty is thus: What happens when the observed labels, $y$, do not match the hidden true labels $y^*$? Our theory concludes that the answer depends on the shape of the noisy distribution $P[Y|X]$ with respect to feature space.

36th Conference on Neural Information Processing Systems (NeurIPS 2022).

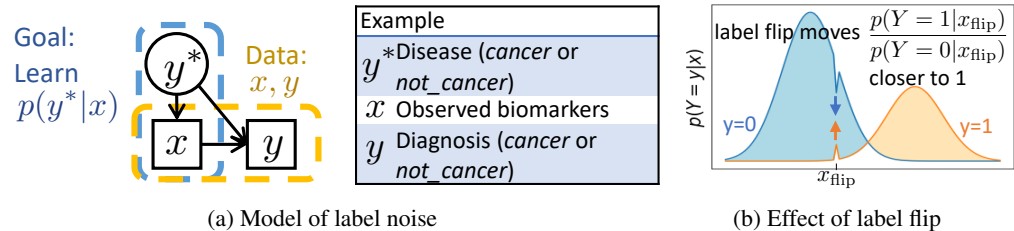

(a) Model of label noise        (b) Effect of label flip

Figure 1: Model of label noise

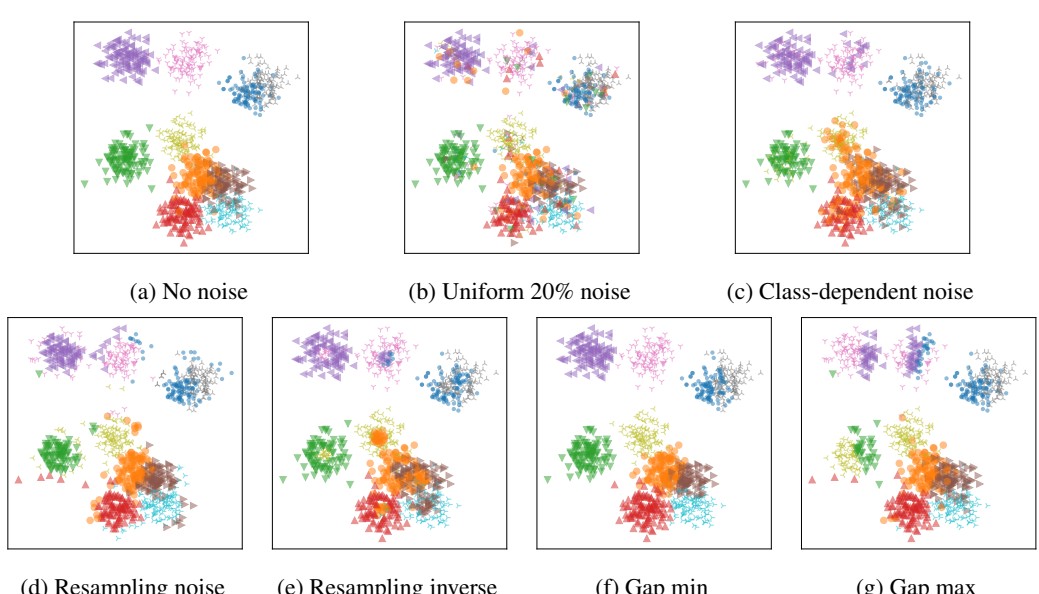

(a) No noise      (b) Uniform 20% noise      (c) Class-dependent noise

(d) Resampling noise    (e) Resampling inverse    (f) Gap min    (g) Gap max

Figure 2: Scatterplot of 2-dimensional 10-class (classes are color and shape coded) Gaussian data. (a) original data without label noise; and with 20% label flip noise for various types of label noise distributions: (b) *uniform*, (c) *class*-dependent, and (d)-(g) *feature*-dependent.

Empirical research suggests that ML is often able to generalize beyond label noise injected in training data to make accurate predictions, particularly when the noisy label is sampled independently of the features and the true label, i.e. $P[Y|X, Y^*] = P[Y]$ [2–5, 22, 24, 25]. This is a remarkable finding considering how messy such data appears, as shown in Fig. 2b. However, this robustness to label noise drops significantly when these approaches test their findings by synthesizing label noise with the assumption that the noise is independent of the features given the true class label, i.e. $P[Y|X, Y^*] = P[Y|Y^*]$. In other words, for a given class, all samples in that class are equally likely to have the same erroneous label, as demonstrated in Fig. 2c. We theoretically analyze the posterior likelihood of the true class given noisy training labels to demonstrate that both of these types of label noise (uniform and class-dependent) behave similarly but with very different *tipping points* of noise level beyond which the true labels are not recoverable.

While the assumption of label noise being independent of features is convenient for generating noisy samples [3, 5, 13, 28], it is not clear that it is an appropriate assumption [4, 24]. More realistically, we expect that label noise occurs when the features are ambiguous; such as when contradictory biomarkers are observed in an oncology patient, or when an image to be classified is blurry. In other words, $P[Y|X, Y^*] \neq P[Y|Y^*]$. Figs. 2d and 2f show data generated with two noise distributions such that label noise is most likely to occur in regions of feature-space that are near class boundaries; while Figs. 2e and 2g show data where label flips are most likely to occur far from true class boundaries. Our theory and empirical work demonstrate that feature-dependent label flips that create noisy class boundaries far from the true class boundaries have the most negative impact on classification accuracy, even at levels of label noise much lower than the tipping points seen in uniform and class-dependent label noise.

We demonstrate theoretically and empirically that classification is generally robust to uniform and class-dependent label noise until the scale of the noise exceeds a threshold that depends on the "spread" of the noise distribution; but that beyond this tipping point, classification accuracy declines rapidly. Yet, we also demonstrate that such robustness to label noise is misleading; because our introduction of feature-dependent label noise shows that classification accuracy can be lowered significantly even for small amounts of label noise. We evaluate, for the first time, the damaging effect of feature-dependent label noise on recent strategies for mitigating label noise. We conclude that the shape of the label noise distribution with respect to feature space is a significant factor in label noise robustness. Our theoretical findings should guide future theoretical and empirical work on label noise to focus studies on the most informative combinations of noise scale and shape; saving significant computational effort while improving the research community's ability to gain insight about how label noise affects learned models.

## 2 Preliminaries

### 2.1 Problem Statement

We consider the problem of label noise in which, for given data sample, $(X, Y^*)$; for some fraction of samples, the true labels $Y^* \in \{1, \ldots, c\}$ are replaced with incorrect (noisy) labels $Y$. The fraction of noisy samples is called the *noise level*. The typical paradigm for modelling such noise is using *label flips* [16, 22], where for each sample, there is some probability that the true label will be flipped to an incorrect label. Which label it flips to is typically modelled in one of two ways: 1) uniform noise (also known as symmetric label noise): each incorrect class is equally likely; or 2) class-dependent noise (also known as asymmetric label noise) in which some classes are more likely than others for a given true class. In the case of class-dependent noise, the noise function is typically represented as a $c \times c$ label transition matrix describing the probably of a transition from $Y^*$ to $Y$. The number of non-zero off-diagonal elements in each row of this matrix (or inversely, the sparsity) characterizes the spread of the noise distribution [13]. In other words, when spread is $c - 1$, the noise distribution is uniform, but the lower the sparsity, the further from uniform the distribution is.

### 2.2 Related Work

Several methods have been developed to improve classifier robustness to noisy labels during training. CleanLab removes noisy samples from training [13]. Methods like CoTeaching weight the samples [5, 21]. Approximate expectation-maximization infers the true label [4]. MentorNet uses a teacher-student learning approach [7]. MixUp interpolates pairs of examples and their labels [28]. SoftLabels treats the labels as learnable parameters [24]. Symmetric cross-entropy (SCE) uses a robust loss function [25]. Similar to our paper, [3] evaluates robustness of existing approaches to label noise. Chen et al. [2] shows that the noisy test accuracy is a quadratic function of the noise level for the case of uniform label noise. Yet, none of these approaches are yet evaluated on feature-dependent noise.

A few papers argue for the need to consider feature-dependent label noise, but they do not evaluate their approaches in this setting [4, 24]. Notably, two papers replace the label-flip paradigm with real-world webly-labeled noise (which is likely feature-dependent) [8, 10], but simultaneously induce a domain shift which makes analysis challenging; and the dependence on the features is unknown.

## 3 Theory

We show that the estimation of the true conditional distribution $P[Y^*|X]$ depends on the level of noise, as well as the shape of the noise. We define the classification problem of $c$ classes with label flips; and examine different types of label noise: uniform, class-dependent, and feature-dependent. Proofs are given in the Supplement.

### 3.1 Problem Definition: Classification with Label Flips

In a classification problem with $c$ classes, we are given an n-sample of pairs $(X, Y^*) \in \mathbb{R}^d \times \{1, \ldots, c\}$, with joint distribution determined by

$$f_k(x) = P[X = dx|Y^* = k], \quad \text{and} \quad P[Y^* = k] = \pi_k^*, \tag{1}$$

for each $k \in \{1, \ldots, c\}$. We want to estimate the *clean posterior*:

$$m_k^*(x) \triangleq P[Y^* = k | X = x] = \frac{\pi_k^* f_k(x)}{\sum_{i=1}^c \pi_i^* f_i(x)}. \tag{2}$$

Rather than observing $Y^*$, we observe the noisy label $Y$ with probability $P[Y = i | Y^* = k, X = x] = \eta_{ki}(x)$ where $\sum_{i=1}^c \eta_{ki}(x) = 1$. In other words, $Y | Y^*, X$ is multinomial distributed such that $1 - \eta_{kk}(x)$ is the probability of a label flip and each $\eta_{ki}(x)$ influences which noisy label, $i$, may be chosen. The impact of label noise is that the estimate of the *noisy posterior* is

$$m_k(x) \triangleq P[Y = k | X = x] = \sum_{i=1}^c \eta_{ik}(x) m_i^*(x) \tag{3}$$

## 3.2 Uniform Label Noise

Classifiers are generally robust to uniform label noise up to a tipping point, as has been well-demonstrated empirically.

**Definition 3.1 (Uniform noise)** *For a classification problem with $c$ classes, given the scale of label noise, $0 \leq \epsilon \leq 1$, the distribution of noisy labels is constant such that:*

$$\eta_{ki} = \begin{cases} 1 - \epsilon, & \text{for } i = k \\ \frac{\epsilon}{c-1}, & \forall i \in [1, \ldots, c] \text{ s.t. } i \neq k \end{cases} \tag{4}$$

Plugging the Definition 3.1 into the noisy posterior Eq (3); the noisy posterior under uniform noise is:

$$m_k(x) = m_k^*(x) - \frac{c\epsilon}{c-1} m_k^*(x) + \frac{\epsilon}{c-1} \tag{5}$$

**Lemma 3.2 (Noisy accuracy)** *Given noisy train and test samples according to Definition 3.1, the noisy test accuracy of a Bayes optimal classifier is quadratic with respect to the noise level $\epsilon$:*

$$\mathbb{E}[I([\arg\max_i m_i(x)] = k) | Y = k] = \bar{m}_k^* \left(1 - \frac{c\epsilon}{c-1}\right)^2 + \frac{\epsilon}{c-1}\left(2 - \frac{c\epsilon}{c-1}\right) \tag{6}$$

**Corollary 3.2.1** *Given noisy train and test samples according to Definition 3.1, the minimum noisy accuracy of $\frac{1}{c}$ occurs at $\epsilon = \frac{c-1}{c}$*

Fig 3a plots the quadratic of Eq (6) which shows that models trained on noisy data will at first decrease in accuracy as tested on noisy labels, but then begin to fit the label noise rather than the true labels as the fraction of noisy labels increases beyond the minimizing point.

**Theorem 3.3 (Clean accuracy)** *Given noisy train samples according to Definition 3.1 and clean test samples, the clean test accuracy of a Bayes optimal classifier is logistic with respect to the noise level $\epsilon$:*

$$\mathbb{E}[I([\arg\max_i m_i(x)] = k) | Y^* = k] \approx \frac{\bar{m}_k^*}{1 + b^{\frac{c}{c-1}(2\bar{m}_k^* - 1)(\epsilon - \frac{c-1}{c})}}, \tag{7}$$

*where $b$ is an arbitrary base of the exponent used in the softmax approximation to the argmax. The larger the value of $b$, the closer to the true maximum function.*

**Corollary 3.3.1** *Given noisy train samples according to Definition 3.1 and clean test samples, clean accuracy drops to $\frac{\bar{m}_k^*}{2}$ at $\epsilon = \frac{c-1}{c}$.*

See Fig 3. For a dataset like ImageNet with 1000 classes, the clean accuracy would only drop off when the noise level is above $99.9\%$. This finding is consistent with empirical studies [17] showing that deep learning models are robust to this type of label noise; particularly when there are large numbers of classes.

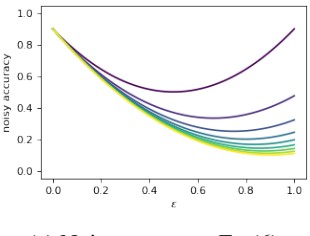
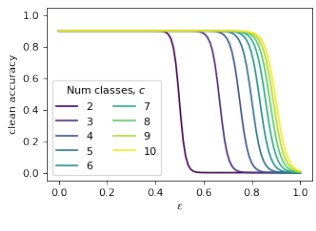

(a) Noisy accuracy, Eq (6)  (b) Clean accuracy, Eq (7)

Figure 3: Theoretical accuracy of predicting noisy or clean labels for uniform label noise (with $m^* = 0.9$ and $b = e^{50}$).

### 3.3 Class-Dependent Flips

We consider the special case of class-dependent label noise that is also well-demonstrated empirically. Often, this class-dependent noise is defined such that $\eta_{ki} = 0$ for several $i$. We refer to the number of non-zero elements as the *spread* of the class-dependent noise function. In the least-spread case, $\eta_{kj} = \epsilon$ for some $j \in \{1, \ldots, c\} \setminus \{k\}$ while $\eta_{ki} = 0$ for all other values of $i \in \{1, \ldots, c\} \setminus \{k, j\}$.

**Definition 3.4 (Class-dependent noise)** *For a classification problem with $c$ classes, given the scale of label noise, $0 \leq \epsilon \leq 1$, and spread of noisy label distribution, $1 \leq s \leq c - 1$; the distribution of noisy labels is constant within each class $k \in \{1, \ldots, c\}$:*

$$\eta_{ki} = \begin{cases} 1 - \epsilon, & for\ i = k \\ \epsilon \cdot t_{ki}, & \forall i \in \{1, \ldots, c\} \setminus \{k\}\ and\ \sum_{i \neq k} |t_{ki}|^0 = s\ and\ \sum_{i \neq k} t_{ki} = 1 \end{cases} \tag{8}$$

Plugging Definition 3.4 into the noisy posterior Eq (3) we see that the noisy posterior under class-dependent noise depends on the spread, $s$, rather than the total number of classes:

$$m_k(x) = m_k^*(x) \left( 1 - \frac{\epsilon(s+1)}{s} \right) + \frac{\epsilon}{s} \tag{9}$$

**Lemma 3.5 (Noisy accuracy)** *Given noisy train and test samples according to Definition 3.4, the noisy test accuracy of a Bayes optimal classifier is quadratic with respect to the noise level $\epsilon$ and spread $s$:*

$$\mathbb{E}[I([\arg\max_i m_i(x)] = k)|Y = k] \quad = \quad \bar{m}_k^* \left( 1 - \frac{\epsilon(s+1)}{s} \right)^2 + \frac{\epsilon}{s} \left( 2 - \frac{\epsilon(s+1)}{s} \right) \tag{10}$$

**Theorem 3.6 (Clean accuracy)** *Given noisy train samples according to Definition 3.4 and clean test samples, the clean test accuracy of a Bayes optimal classifier is logistic with respect to the noise level $\epsilon$:*

$$\mathbb{E}[I([\arg\max_i m_i(x)] = k)|Y^* = k] \quad \approx \quad \frac{\bar{m}_k^*}{1 + b^{\frac{s+1}{s}(2\bar{m}_k^* - 1)(\epsilon - \frac{s}{s+1})}} \tag{11}$$

Class-dependent noise impacts classification accuracy in a similar way to uniform noise; however, the tipping-point in class-dependent noise is controlled by the spread rather than by the total number of classes. Thus, even for a large number of classes, the tipping point can be as low as $\epsilon = 1/2$.

### 3.4 Feature-Dependent Label Noise

We consider the more general paradigm of feature-dependent label noise which has not received much attention either theoretically or empirically. Consider the simplest feature-dependent label-flip scenario where a single sample $(X = x, Y^* = k)$ for some fixed $x$ with true label $k$ will have noise introduced as $(X = x, Y = j)$ for some $j \in \{1, \ldots, c\}$. No matter which noisy label $j$ appears, the posterior likelihood of the noisy data will be reduced to $m_k(x) = (1 - p(x))m_k^*(x)$; while the

posterior likelihood of the incorrect label will increase to $m_j(x) = m_j^*(x) + p(x)m_k^*(x)$. In the Bayes optimal classifier, classification errors occur when:

$$m_k(x) < \max_{i \in \{1,\ldots,c\}\setminus k} m_i(x) \quad , \tag{12}$$

for true label $k$ for given point $x$. Therefore, for a fixed $x$, the label-flip $Y = j$ that would have the most impact on classification (the greatest increase of the right-hand side of the inequality) is

$$j = \arg \max_{i \in \{1,\ldots,c\}\setminus k} m_i(x) \quad . \tag{13}$$

An immediate consequence of this observation is that worst-case label noise is a function of $x$; therefore any evaluation of label noise (empirical or theoretical) that does not depend on features is optimistic.

Now, consider which samples, i.e. the shape of $\eta_{kk}(x)$, would have the most impact on clean classification accuracy. By re-arranging Eq (12), we see that the ratio $m_k(x)/\max_{i\in\{1,\ldots,c\}\setminus k} m_i(x)$ defines the optimal classification boundaries. By targeting label flips on samples $x$ where this ratio is high, a new (noisy) classification boundary can be created; as illustrated in Fig. 1b.

**Claim 3.7 (Worst-case noise)** *For a classification problem with c classes, the samples, x, that have the greatest impact on clean label prediction are those that maximize the ratio between the likelihood of the true class to the likelihood of the most-likely incorrect class:*

$$\eta_{ki}(x) \propto \begin{cases} \left(\frac{P[Y^*=k|x]}{\max_{j\neq k} P[Y^*=j|x]}\right)^{-1}, & \text{for } i = k \\ I[j = \arg\max_{i\neq k} P[Y^* = i|x]], & \forall i \in \{1,\ldots,c\} \setminus \{k\} \end{cases} \tag{14}$$

### 3.5 Discussion of theoretical results

**Relationship to existing theory** It is known that label noise is not generally distinguishable from other label uncertainty (such as class overlap) [19, 20]. Various approaches provide guarantees of label noise identification through constraints on $m_k^*(x)$ including "purity" of classes [1, 11, 15, 19, 20]. Other methods provide guarantees on robust learning given constraints on the level of label noise [1, 4, 26], or level of noise in classification [2, 13], or only for binary classification [12], or restricted to generalized linear models [21]. In contrast, we consider the unconstrained problem to analyze the effect of label flips for any noise level and feature-dependent noise. Rather than proving guarantees about effectiveness of models under constraints; we show that when such constraints do not hold, classification degrades suddenly—not gracefully.

Setting $\bar{m}_k^* = 1$ in our Corollary 3.2.1 yields the special case given in Corollary 1.1 of [2]. Our generalization beyond the perfect classifiers (defined in our notation as $m_k^*(x) = \bar{m}_k^* = 1$) assumed in [2] is important because if we apply our result to the problem of identifying noisy labels (as presented in [2]); then we see that the accuracy of predicting noisy labels is a function of $m_k^*(x)$ — i.e. accurate prediction of label noise *depends* on the features — even when label noise is *independent* of the features.

**Implications** The standard assumption of label noise being independent of features can lead to overly-optimistic findings on the robustness of classification algorithms. We can see that if label noise is applied uniformly randomly in feature space (even if it is non-uniform across classes), then even as the noisy likelihood begins to decrease, classification accuracy remains high until the likelihood estimate decreases below the tipping point.

## 4 Synthetic Data Experiments

The theory is validated empirically on synthetic data for which $P[Y^*|X]$ and $P[Y|X]$ are known. Data is 2-d Gaussian with up to 10 classes, with some classes overlapping and others easily-separable as shown in Fig. 2a. We also empirically evaluate 5-d Gaussian data with up to 50 classes. All code will be made available as open-source.

For uniform noise, we sample $y \sim \eta_{ki}$ according to Eq. 4 for varying numbers of classes, $c \in \{2, 4, 6, 8, 9, 10\}$, and noise ratio, $\epsilon \in \{0, 0.1, \ldots, 0.9, 1\}$. For class-dependent noise, we sample

| Noise name | $\eta_{kk}(x)$ | $\eta_{kj}(x)$ |
|---|---|---|
| Uniform-x | $= 1 - \epsilon$ | $\propto P[Y^* = j|x]$ |
| Resampling | $\propto P[Y^* = k|x]$ | $\propto P[Y^* = j|x]$ |
| InverseResampling | $\propto P[Y^* = k|x]^{-1}$ | $\propto P[Y^* = j|x]$ |
| GapMin | $\propto \frac{P[Y^*=k|x]}{\max_{j\neq k} P[Y^*=j|x]}$ | $= \epsilon I[j == \arg\max_{i\neq k} P[Y^* = i|x]]$ |
| GapMax | $\propto \left(\frac{P[Y^*=k|x]}{\max_{j\neq k} P[Y^*=j|x]}\right)^{-1}$ | $= \epsilon I[j == \arg\max_{i\neq k} P[Y^* = i|x]]$ |

Table 1: Types of feature-dependent noise.

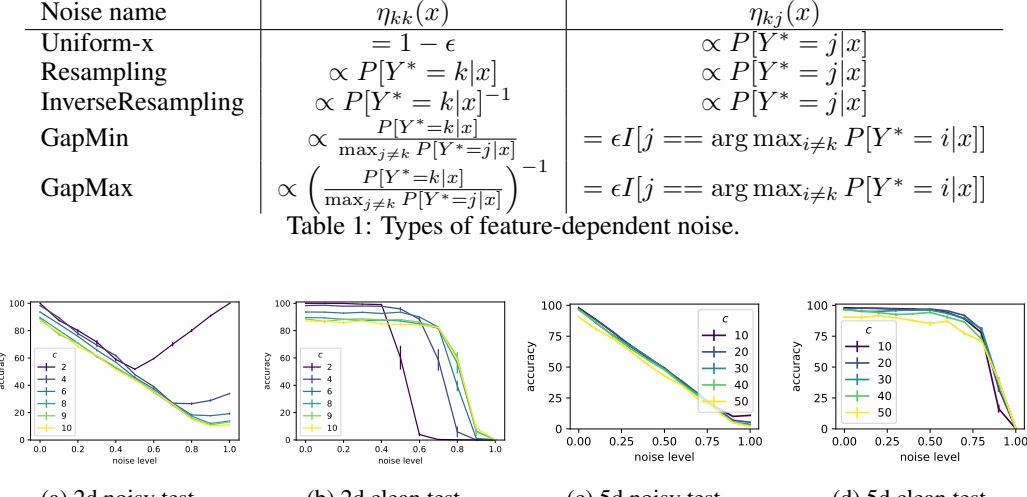

(a) 2d noisy test     (b) 2d clean test     (c) 5d noisy test     (d) 5d clean test

Figure 4: Deep learning results on synthetic data with uniform noise.

$y \sim \eta_{ki}$ according to Eq. 8 for varying numbers of classes, $c \in \{2, 4, 6, 8, 9, 10\}$; noise ratio, $\epsilon \in \{0, 0.1, \ldots, 0.9, 1\}$; and spread $\in \{1, 2, 5\}$. For feature-dependent noise, we sample $y$ in five different ways as summarized in Table 1. (1) For *uniform-x*, the probability of a given sample flipping is constant: $\eta_{kk}(x) = 1 - \epsilon$; while the choice of label to flip to is proportional to the most-likely incorrect label: $\eta_{kj}(x) \propto P[Y^* = j|x]$. (2) For *resampling* noise, $\eta_{kk}(x) \propto P[Y^* = k|x]$, in other words, samples that are least likely in a true class are the most likely to flip. (3) For *inverse resampling*, $\eta_{kk}(x) \propto P[Y^* = k|x]^{-1}$, i.e., samples with the highest likelihood in the true class are most likely to flip. (4) For *minimum-gap* noise, $\eta_{kk}(x) \propto \frac{P[Y^*=k|x]}{\max_{j\neq k} P[Y^*=j|x]}$, so that the samples nearest decision boundaries are the most likely to flip. (5) For *maximum-gap* noise, $\eta_{kk}(x) \propto \frac{\max_{j\neq k} P[Y^*=j|x]}{P[Y^*=k|x]}$, so that the samples farthest from decision boundaries are the most likely to flip.

There are 100 samples per class in the training set and 100 samples per class in the test set. The feature matrices $\mathbf{X}_{\text{train}}$ and $\mathbf{X}_{\text{test}}$ are the same across all trials. The corresponding noisy label vectors $\mathbf{Y}_{\text{train}}$ and $\mathbf{Y}_{\text{test}}$ are sampled for each combination of noise settings. We also evaluate the accuracy of the held-out clean test labels $\mathbf{Y}_{\text{test}}^*$. The noise level $\epsilon$ varies from 0 to 1 in 0.1 increments. A neural network with 2 hidden layers is trained; and further details of the architecture is given in the Supplement. The model is trained 5 times starting from a different random seed; with the mean and standard deviation of the accuracies reported.

**Results** Affirming existing results [2, 17, 22], and our theoretical analysis, the similarity of Figure 4 to the theoretical expectation of Fig. 3 is striking. The neural network is robust to uniform label noise up to a point, where that tipping point depends on the number of classes. Figure 4 gives insight into what is happening during training. We see by the very low accuracies on the noisy test labels that the model is not able to fit the noise from this uninformative noise distribution. Yet, there is enough signal still in the data to learn the patterns that generalize to the clean test set. Beyond the tipping

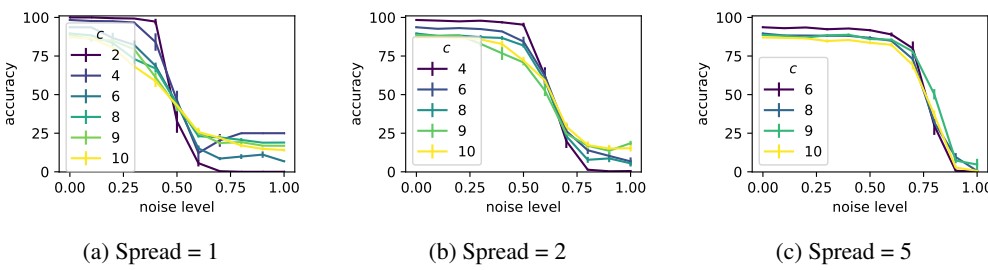

(a) Spread = 1     (b) Spread = 2     (c) Spread = 5

Figure 5: Deep learning results on synthetic 2D data with class-dependent noise.

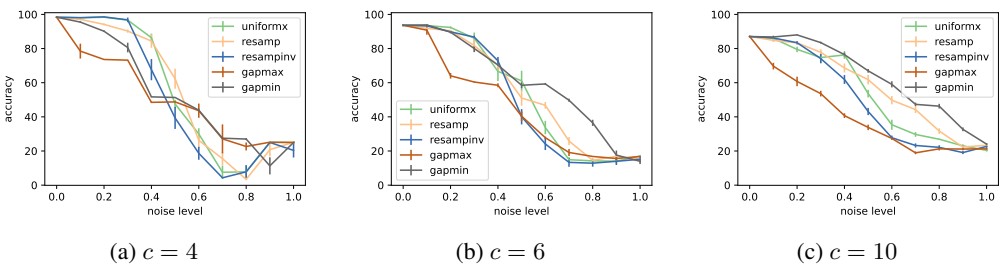

(a) $c = 4$             (b) $c = 6$             (c) $c = 10$

Figure 6: Deep learning results on synthetic 2D data with feature-dependent noise.

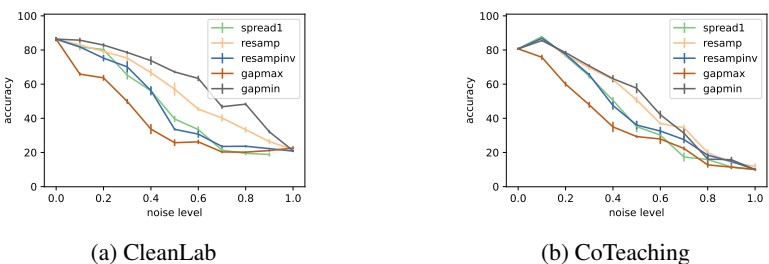

(a) CleanLab             (b) CoTeaching

Figure 7: Label noise mitigation methods on synthetic 2D data with 10 classes.

point (the minimizing point in the quadratic function of the noisy accuracy), the neural network begins fitting the noise instead of the true signal, and the clean accuracy drops off steeply.

Results on class-dependent noise as shown in Figure 5, also affirm existing results [13, 25, 28] and our theoretical analysis. We see that the tipping point depends on the *spread* rather than the total number of classes in the classification task. The most harmful class-dependent noise occurs with spread=1, that is when labels from a given class flip to one specific erroneous label, the tipping point occurs when the noise level is $0.5$.

With the introduction of feature-dependent label noise as seen in Fig. 6, we see that the neural network is not always robust even at low levels of label noise. In particular, the worst-case label noise, GapMax, of Claim 3.7 significantly lowers classification accuracy even at $\epsilon = 0.1$ and $0.2$. Interestingly, the classification accuracy from uniform-x noise is statistically indistinguishable from class-dependent noise; indicating that the dependence of $\eta_{kk}(x)$ on $x$ is important.

Fig. 7 highlights the clean test accuracy of two label-noise mitigation methods: CleanLab [13] and CoTeaching [5]. More methods are detailed in the Supplement. At $\epsilon = 0.1$ both strategies improve results compard with the baseline of Fig. 6c. But for higher levels of $\epsilon$, CleanLab has little effect while CoTeaching performs worse than baseline.

Table 2: CIFAR10 clean accuracy. Highest accuracy for each noise setting is highlighted.

| Class noise | $\epsilon = 0.2$ | | | $\epsilon = 0.4$ | | | $\epsilon = 0.6$ | | | $\epsilon = 0.8$ | | |
|---|---|---|---|---|---|---|---|---|---|---|---|---|
| | $s = 1$ | $s = 2$ | $s = 5$ | $s = 1$ | $s = 2$ | $s = 5$ | $s = 1$ | $s = 2$ | $s = 5$ | $s = 1$ | $s = 2$ | $s = 5$ |
| Baseline | 78.15 | 80.01 | 80.80 | 58.94 | 63.49 | 66.88 | 35.26 | 40.62 | 48.70 | **15.07** | **15.28** | 19.44 |
| CleanLab | 88.41 | 88.89 | 88.87 | **83.39** | 86.27 | **86.66** | 21.74 | **73.06** | **81.73** | 4.51 | 7.96 | **50.69** |
| MixUp | **90.62** | **90.45** | **90.69** | 77.77 | **86.33** | 85.01 | 12.27 | 64.29 | 76.02 | 5.16 | 4.59 | 28.16 |
| Co-Teaching | 87.23 | 88.12 | 88.09 | 67.79 | 71.31 | 71.21 | 22.80 | 43.33 | 50.17 | 6.19 | 8.91 | 22.22 |
| SCE | 86.21 | 86.62 | 87.38 | 57.42 | 82.71 | 84.23 | 15.53 | 42.86 | 63.47 | 10.68 | 6.26 | 15.90 |
| **Feature noise** | | | | | | | | | | | | |
| Baseline | 76.92 | 77.43 | 78.73 | 59.86 | 60.84 | 63.72 | 35.89 | 39.49 | 40.70 | 16.81 | 16.87 | 17.55 |
| CleanLab | 88.56 | 88.47 | 88.55 | **82.47** | **84.30** | **86.03** | 32.06 | **56.90** | **76.04** | **22.10** | **22.58** | **26.37** |
| MixUp | **88.70** | **89.85** | **89.83** | 76.14 | 78.30 | 82.48 | **41.75** | 53.59 | 60.15 | 10.96 | 13.08 | 25.43 |
| Co-Teaching | 87.15 | 87.24 | 88.01 | 64.04 | 67.77 | 70.50 | 28.76 | 35.09 | 46.70 | 12.41 | 14.23 | 17.68 |
| SCE | 87.26 | 87.65 | 87.39 | 74.71 | 77.70 | 83.68 | 23.97 | 33.65 | 66.98 | 18.81 | 17.03 | 15.52 |

Table 3: CIFAR100 clean accuracy. Highest accuracy for each noise setting is highlighted.

| Class noise | $\epsilon = 0.2$ | | | $\epsilon = 0.4$ | | | $\epsilon = 0.6$ | | | $\epsilon = 0.8$ | | |
|---|---|---|---|---|---|---|---|---|---|---|---|---|
| | $s=1$ | $s=2$ | $s=5$ | $s=1$ | $s=2$ | $s=5$ | $s=1$ | $s=2$ | $s=5$ | $s=1$ | $s=2$ | $s=5$ |
| Baseline | 57.44 | 57.20 | 57.51 | 42.24 | 45.49 | 47.81 | **24.32** | 29.43 | 34.17 | **10.37** | **10.63** | 15.24 |
| CleanLab | 51.27 | 52.79 | 53.59 | 32.84 | 40.95 | 45.06 | 9.11 | 16.36 | 30.30 | 3.83 | 4.10 | 6.60 |
| MixUp | **64.79** | **65.18** | **65.68** | **50.65** | **57.35** | **58.27** | 18.64 | **34.07** | **48.23** | 5.00 | 7.58 | **17.93** |
| SCE | 54.88 | 56.18 | 54.93 | 33.72 | 46.87 | 50.54 | 12.59 | 19.96 | 36.70 | 6.26 | 8.72 | 12.16 |
| Co-Teaching | 55.85 | 58.66 | 59.65 | 37.79 | 41.83 | 45.29 | 21.11 | 24.43 | 28.00 | 6.72 | 9.28 | 11.57 |
| **Feature noise** | | | | | | | | | | | | |
| Baseline | 53.80 | 54.13 | 54.64 | 37.17 | 38.73 | 39.34 | 20.38 | 21.00 | 22.71 | 7.94 | 8.38 | 8.72 |
| CleanLab | 51.74 | 52.75 | 53.53 | 33.42 | 36.36 | 40.58 | 11.91 | 14.64 | 22.56 | 6.56 | 6.86 | 8.04 |
| MixUp | **63.10** | **62.67** | **63.42** | **47.73** | **50.34** | **51.87** | **27.06** | **29.86** | **33.76** | **9.16** | **10.55** | **13.28** |
| SCE | 53.43 | 54.02 | 55.51 | 32.43 | 36.66 | 42.39 | 9.42 | 11.70 | 17.61 | 7.29 | 7.41 | 7.76 |
| Co-Teaching | 58.40 | 58.16 | 59.28 | 35.71 | 38.93 | 40.80 | 14.12 | 14.40 | 17.05 | 6.17 | 6.28 | 6.08 |

# 5 Image Benchmark Experiments

We compare methods for learning with noisy labels, including CleanLab with the confident joint [13], MixUp [28], symmetric cross entropy (SCE) [3], and CoTeaching [5]; using the public-license open-source code provided by the authors. For all methods the base architecture is ResNet-32 [6] with more details including computational costs, and extended empirical results, in the Supplement. We also compare the baseline performance which is the ResNet-32 architecture without any label noise mitigation strategy.

We use the classification benchmarks CIFAR-10 and CIFAR-100 of 32x32-pixel color images in 10 or 100 classes, with 60,000 images per dataset [9]. Class-dependent noisy labels $Y|Y^*$ are generated according to Definition 3.4, assuming that the labels given in the dataset are the true labels $Y^*$.

**Generating feature-dependent label noise**   To generate label noise $Y|X$ that is dependent on features but independent from the learned classification function itself, we use similarity-preserving image hashes [14, 27]. For each image in the dataset, we compute the similarity-preserving hash, $h(x)$; which puts visually-similar (rather than semantically-similar) images near each other. Then for each class, we calculate the average hash, $\bar{h}_k$, which represents the "center" of the class. For class-conditional noise, we generate label flips between classes that are $s$-nearest neighbors based on the hash centers and spread $s$. For feature-dependent noise, we generate label flips using the GapMax function of Table 1 based on the distance from a particular sample's hash to the $s$-nearest neighbor hash centers, i.e. $P[Y^* = k|x] \propto 1/\delta_k(x)$ where $\delta_k(x) = \|h(x) - \bar{h}_k\|_2$. We investigate two different similarity-preserving hashes: 1) VisHash [14] which was designed to detect near-copy images ignoring changes in color and shading, so it will put images with similar shapes near each other; and 2) $h(x)$ is a ResNet-50 [6] pre-trained on ImageNet-1000 [18] using the activations of the layer before the fully connected layers as the hash vector. The results for the VisHash-derived label flips are provided here with the ResNet-50 driven label flips provided in the Supplement, noting that the general trends are similar for these two types of feature-driven label flips.

**Results**   The results for class-dependent label flips on CIFAR-10 in Table 2 show general patterns of what our theory predicts: accuracy drops severely between $\epsilon = 0.4$ and $\epsilon = 0.6$ when $s = 1$. For $s = 2$, the drop is less severe until $\epsilon = 0.8$. Also supporting our theory, the results for feature-dependent label flips show more gradual decline in accuracy so that on average the accuracy is lower for feature-dependent label flips than for class-dependent label flips when $\epsilon \in \{0.2, 0.4\}$, as well as for higher values of $\epsilon$ when spread, $s \in \{2, 5\}$. Table 3 shows that similar trends hold for a dataset with a much high number of classes. It is interesting to note that when a mitigation strategy is performing poorly (e.g. SCE on CIFAR10 at $\epsilon = 0.2$ or CleanLab on CIFAR100 at $\epsilon = 0.2$) compared with the other methods; then it does not seem to matter whether the noise is feature-dependent or class-dependent. Therefore, we see that in general the robustness of classifiers with successful label noise mitigation strategies behave differently whether that noise is feature-dependent. Feature-dependent noise reduces classification accuracy at lower levels of noise.

CleanLab and MixUp show the strongest results, but there is no clear winner among mitigation strategies for label noise; as all methods display weaknesses under various noise levels or noise types. One promising result is that the mitigation strategies tend to narrow the gap in performance between

handling class noise and feature noise; which is consistent with our findings on the synthetic data in Figure 7 . The strong performance of MixUp on CIFAR-100 is probably due to it being particularly well-suited to the hierarchical nature of the classes [23, 28]. Whereas, CleanLab struggles with CIFAR-100 probably because the low baseline accuracy even at low levels of noise makes it difficult to confidently predict clean versus noisy samples.

## 6   Conclusions

**Limitations**   Our theoretical results are limited to the Bayes optimal classifier. As with most theory, the degree to which these results apply in practice must be explored empirically.

**Future Directions**   The theory could be extended to explain why learning from massive, noisy datasets can be so successful, especially when the noisy dataset is used as a pre-training task: major patterns in data can be learned despite label noise as long as there is some signal within the noise. Our theory could also be used to guide the curation of datasets as they are being collected. Not all labels need to be clean, but enough labels in every class across the relevant feature space do need to be clean. Data-specific approaches may look for regions of feature space where labels tend to be noisy and focus more effort on cleaning labels for those regions.

**Conclusion**   In both the uniform and class-dependent label flip scenarios, classification is robust to label noise up to a tipping point; and beyond that tipping point, classification fails catastrophically. These theoretical findings are helpful in designing evaluation approaches for label noise detection and mitigation. Considering that label noise mitigation strategies can require training 2 models (such as for CoTeaching) or even 6 models (such as for CleanLab); we can see why empirical studies are computationally limited to investigating just a few variations on noise level and shape. We now know that levels of noise near where the tipping-point occurs will be most informative; that there is little reason to evaluate various levels of spread of class-dependent noise, and that incorporating feature-dependent noise is most informative for practical applications.

Classification accuracy is significantly lower at small levels of noise when the noise distribution depends on features and is targeted at samples with the highest gap between the likelihood of the true label and the next-most-likely label. While various label noise mitigation strategies provide guarantees about conditions under which they work; we see that the mitigation strategies on real data generally fail for similar scale and shape of label noise distributions. Realistically, we expect that label noise does depend on the features; and this paper demonstrates theoretically and empirically that the old assumption of label noise—being independent of features given the class—leads to an overly optimistic expectation of classifier robustness in the face of label noise. While it is challenging to model feature-dependent label noise; it is clearly necessary as we see significantly lower classification accuracy when label noise is feature-dependent.

## Acknowledgments and Disclosure of Funding

Research presented in this paper was supported by the Laboratory Directed Research and Development program of Los Alamos National Laboratory under project number 20210043DR.

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
