# Supplement: Robustness to Label Noise Depends on the Shape of the Noise Distribution

**Diane Oyen**
Los Alamos National Lab
doyen@lanl.gov

**Michal Kucer**
Los Alamos National Lab

**Nick Hengartner**
Los Alamos National Lab

**Har Simrat Singh**
Los Alamos National Lab


# A Theory: Proofs

Proofs provided for theoretical results of Section 3.

## A.1 Uniform noise

*Proof.* Lemma 3.2:
Let $c$, $\epsilon$, $m_k^*(x)$, and $m_k(x)$ by defined as in Eq 2, Eq 3, and Definition 3.1. So,

$$\mathbb{E}[I([\arg\max_i m_i(x)] = k)|Y = k] =$$

$$= \sum_{i=1}^{c} m_i(x)P[Y^* = i|Y = k]$$

$$= m_k(x)P[Y^* = k|Y = k] + \sum_{i \neq k} m_i(x)P[Y^* = i|Y = k]$$

$$= m_k(x)(1 - \epsilon) + \frac{\epsilon}{c-1}\sum_{i \neq k} m_i(x) \qquad \text{(Definition 3.1 Uniform noise)}$$

$$= m_k(x)(1 - \epsilon) + \frac{\epsilon}{c-1}(1 - m_k(x)) \qquad (\sum_{i=1}^{c} m_i(x) = 1)$$

$$= \frac{\epsilon}{c-1} + m_k(x)\left(1 - \frac{c\epsilon}{c-1}\right)$$

$$= \frac{\epsilon}{c-1} + \left(m_k^*(x) - \frac{c\epsilon}{c-1}m_k^*(x) + \frac{\epsilon}{c-1}\right)\left(1 - \frac{c\epsilon}{c-1}\right) \qquad \text{(Substitute } m_k(x) \text{ with Eq 3)}$$

$$= \frac{\epsilon}{c-1} + m_k^*(x) - \frac{c\epsilon}{c-1}m_k^*(x) + \frac{\epsilon}{c-1} - \frac{c\epsilon}{c-1}m_k^*(x) + \frac{c^2\epsilon^2}{(c-1)^2}m_k^*(x) - \frac{c\epsilon^2}{(c-1)^2}$$

$$= m_k^*(x)\left(1 - \frac{2c\epsilon}{c-1} + \frac{c^2\epsilon^2}{(c-1)^2}\right) + \frac{2\epsilon}{c-1} - \frac{\epsilon^2 + c\epsilon^2 - \epsilon^2}{(c-1)^2}$$

$$= m_k^*(x)\left(1 - \frac{c\epsilon}{c-1}\right)^2 + \frac{\epsilon}{c-1}\left(2 - \frac{c\epsilon}{c-1}\right) \qquad \square$$

*Proof.* Corollary 3.2.1:

Minimize Eq 6 with respect to $\epsilon$:

$$0 = \frac{d}{d\epsilon}\left[\bar{m}_k^*\left(1 - \frac{c\epsilon}{c-1}\right)^2 + \frac{\epsilon}{c-1}\left(2 - \frac{c\epsilon}{c-1}\right)\right]$$

$$0 = \frac{d}{d\epsilon}\left[\bar{m}_k^* - \frac{2c\epsilon}{c-1}\bar{m}_k^* + \frac{c^2\epsilon^2}{(c-1)^2}\bar{m}_k^* + \frac{2\epsilon}{c-1} - \frac{c\epsilon^2}{(c-1)^2}\right]$$

$$0 = -\frac{2c}{c-1}\bar{m}_k^* + \frac{2c^2\epsilon}{(c-1)^2}\bar{m}_k^* + \frac{2}{c-1} - \frac{2c\epsilon}{(c-1)^2}$$

$$0 = -c^2\bar{m}_k^* + c\bar{m}_k^* + c^2\epsilon\bar{m}_k^* + c - 1 - c\epsilon \qquad \text{(Multiply by } \frac{(c-1)^2}{2}\text{)}$$

$$c\epsilon(1 - c\bar{m}_k^*) = (1 - c\bar{m}_k^*)(c-1)$$

$$\epsilon = \frac{(1 - c\bar{m}_k^*)(c-1)}{c(1 - c\bar{m}_k^*)}$$

$$\epsilon = \frac{(c-1)}{c} \qquad \text{(for } \bar{m}_k^* \neq \frac{1}{c}\text{)}$$

The 2nd-derivative is strictly positive as long as $\bar{m}_k^* > \frac{1}{c}$. Assert:

$$\frac{d}{d\epsilon}\left[-\frac{2c}{c-1}\bar{m}_k^* + \frac{2c^2\epsilon}{(c-1)^2}\bar{m}_k^* + \frac{2}{c-1} - \frac{2c\epsilon}{(c-1)^2}\right] > 0$$

$$\frac{2c^2}{(c-1)^2}\bar{m}_k^* - \frac{2c}{(c-1)^2} > 0$$

$$\frac{2c^2}{(c-1)^2}\bar{m}_k^* > \frac{2c}{(c-1)^2}$$

$$\bar{m}_k^* > \frac{1}{c}$$

therefore setting $\epsilon = \frac{c-1}{c}$ gives the global minimum when $\bar{m}_k^* > \frac{1}{c}$. □

Note that the condition that $\bar{m}_k^* > \frac{1}{c}$ is not necessary for the proof, but it makes the results sensible while simplifying the proof. In the case that $\bar{m}_k^* = \frac{1}{c}$, i.e. classification is no better or worse than random chance; Eq 6 is constant and so $\epsilon = \frac{c-1}{c}$ trivially minimizes Eq 6. In the case that $\bar{m}_k^* < \frac{1}{c}$, i.e. classification is worse than random chance; then we get a vacuous result in which label noise actually improves classification and $\epsilon = \frac{c-1}{c}$ maximizes Eq 6.

*Proof.* Theorem 3.3:

$$\mathbb{E}[I([\arg\max_i m_i(x)] = k)|Y^* = k] = P[Y^* = k|X = x] \cdot I([\arg\max_i m_i(x)] = k)$$

$$\approx m_k^*(x)\frac{b^{m_k(x)}}{\sum_{i=1}^c b^{m_i(x)}}$$

$$= m_k^*(x)\frac{b^{m_k(x)}}{b^{m_k(x)} + \sum_{i\neq k} b^{m_i(x)}}$$

$$= \frac{m_k^*(x)}{1 + \sum_{i\neq k} b^{(m_i(x) - m_k(x))}}$$

Let $u = m_i(x) - m_k(x)$. So,

$$
\begin{aligned}
u &= m_i(x) - m_k(x) \\
&= m_i^*(x) - \frac{c\epsilon}{c-1}m_i^*(x) + \frac{\epsilon}{c-1} - m_k^*(x) + \frac{c\epsilon}{c-1}m_k^*(x) - \frac{\epsilon}{c-1} \quad &\text{(Equation 5)} \\
&= m_i^*(x) - m_k^*(x) + \frac{c\epsilon}{c-1}(m_k^*(x) - m_i^*(x)) \\
&= (m_k^*(x) - m_i^*(x))\left(\frac{c\epsilon}{c-1} - 1\right) \\
&\leq (2m_k^*(x) - 1)\left(\frac{c\epsilon}{c-1} - 1\right) \quad &(m_i^*(x) \leq 1 - m_k^*(x)) \\
&= \frac{c}{c-1}(2m_k^*(x) - 1)\left(\epsilon - \frac{c-1}{c}\right)
\end{aligned}
$$

Plug $u$ back in:

$$
\begin{aligned}
\mathbb{E}[I([\arg\max_i m_i(x)] = k)|Y^* = k] &\approx \frac{m_k^*(x)}{1 + \sum_{i \neq k} b^u} \\
&\approx \frac{m_k^*(x)}{1 + b^{\frac{c}{c-1}(2m_k^*(x)-1)(\epsilon - \frac{c-1}{c})}} \quad ,
\end{aligned}
$$

$\square$

## A.2 Class-dependent noise

Lemma 3.5 follows from Definition 3.4 and the noisy posterior likelihood given in Eq 9.

*Proof.* Lemma 3.5: Let $c$, $\epsilon$, $m_k^*(x)$, and $m_k(x)$ by defined as in Eq 2, Eq 3, and Definition 3.4. So,

$$
\begin{aligned}
\mathbb{E}[I([\arg\max_i m_i(x)] = k)|Y = k] &= \\
&= \sum_{i=1}^c m_i(x)P[Y^* = i|Y = k] \\
&= m_k(x)P[Y^* = k|Y = k] + \sum_{i \neq k} m_i(x)P[Y^* = i|Y = k] \\
&\approx m_k(x)(1 - \epsilon) + \frac{\epsilon}{s}\sum_{i \neq k; t_{ki} > 0} m_i(x) \quad &\text{(Definition 3.4)} \\
&\leq m_k(x)(1 - \epsilon) + \frac{\epsilon}{s}(1 - m_k(x)) \quad &(\textstyle\sum_{i \neq k; t_{ki} > 0} m_i(x) \leq 1 - m_k(x)) \\
&= \frac{\epsilon}{s} + m_k(x)\left(1 - \frac{(s+1)\epsilon}{s}\right) \\
&= \frac{\epsilon}{s} + \left(m_k^*(x) - \frac{(s+1)\epsilon}{s}m_k^*(x) + \frac{\epsilon}{s}\right)\left(1 - \frac{(s+1)\epsilon}{s}\right) \quad &\text{(Substitute } m_k(x) \text{ with Eq 9)} \\
&= m_k^*(x)\left(1 - \frac{(s+1)\epsilon}{s}\right)^2 + \frac{\epsilon}{s}\left(2 - \frac{(s+1)\epsilon}{s}\right) \quad &\square
\end{aligned}
$$

The proof of Theorem 3.6 is identical to that of Theorem 3.3 except using the value of $m_k(x)$ given in Eq 9 instead of the value given in Eq 5; or equivalently by simply replacing $c$ in Theorem 3.3 with $s + 1$ to yield Theorem 3.6.

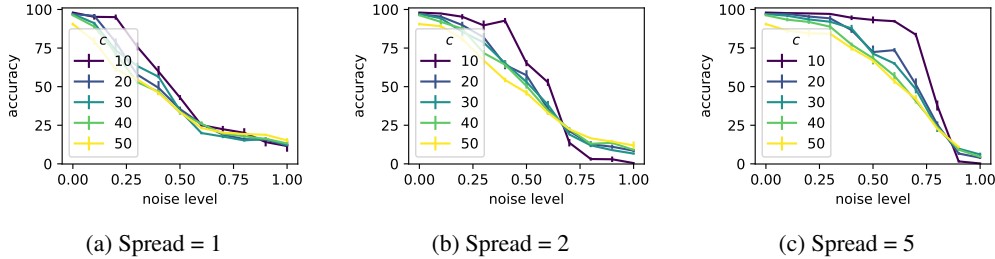

(a) Spread = 1  (b) Spread = 2  (c) Spread = 5

Figure S1: Deep learning results on synthetic 5D data with class-dependent noise.

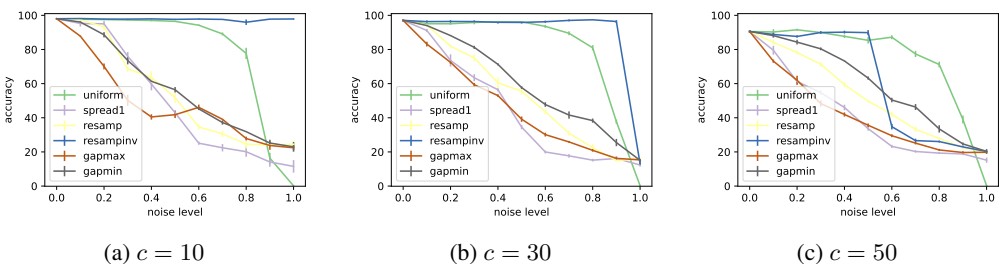

(a) $c = 10$  (b) $c = 30$  (c) $c = 50$

Figure S2: Deep learning results on synthetic 5D data with feature-dependent noise.

# B  Further Synthetic Data Experiments

## B.1  Learning details

All of our code and experiments are implemented in Python using the PyTorch [5] deep learning framework. Synthetic data experiments are performed on an internal cluster where each server node is equipped with 4 NVIDIA 1080Ti GPUs, paired with a 24-core Intel Broadwell E5-2650 v4 CPU and 125GB of memory. All of these experiments use a simple neural network architecture with 2 fully-connected hidden layers of width 10 each, with tanh activation functions and softmax final layer. We use cross-entropy loss with the Adam optimizer with learning rate 0.001 [2]. Various settings for the architecture, loss function, and optimizer were evaluated on clean data (no label noise) to determine the best

For each noise setting, it takes about 70 seconds on a single GPU to learn 5 randomly-initialized models. Therefore, the total computation time for the synthetic data experiments is about 24 compute hours (8 noise types times 10 noise levels times 16 datasets times 70 seconds). The mitigation strategies take longer because they require a 5-fold cross validation (for CleanLab) or training dual networks (for CoTeaching). Therefore, we only evaluate these on a subset of noise types (5 noise types), and numbers of classes/dimensions (2 datasets) for a compute time of about 21 hours for CleanLab and 11 hours for CoTeaching.

## B.2  Results on 5d Data

Figure S1 shows that the we see the same result on class-dependent noise for 5-dimensional data as already shown in 2-dimensional data. Accuracy on data trained with class-dependent noise depends on the level of noise $\epsilon$, as well as the spread, $s$; but it does not depend directly on the number of classes, $c$.

Figure S2 shows that the we see similar trends on feature-dependent noise for 5-dimensional data as already shown in 2-dimensional data. Accuracy on data trained with gap-max feature-dependent noise performs worse than other types of noise at low levels, $\epsilon$, of noise. Interestingly, the difference between feature-dependent and class-dependent noise appears to diminish for an increased number of classes.

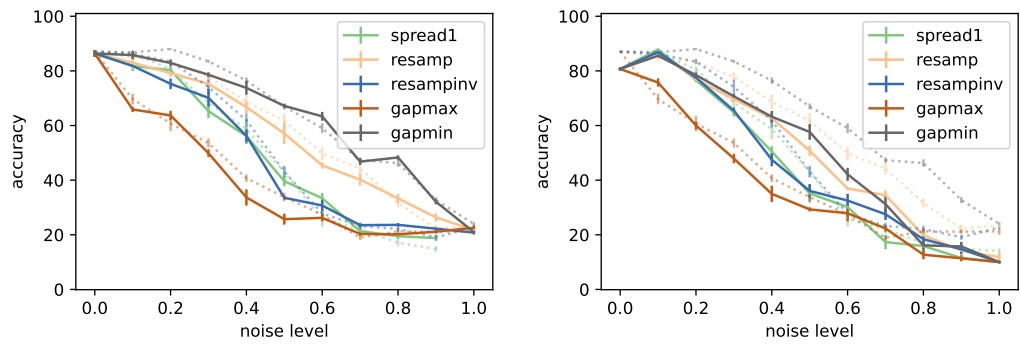

(a) CleanLab (solid) against baseline (dashed)       (b) CoTeaching(solid) against baseline (dashed)

Figure S3: Label noise mitigation methods on synthetic 2D data with 10 classes.

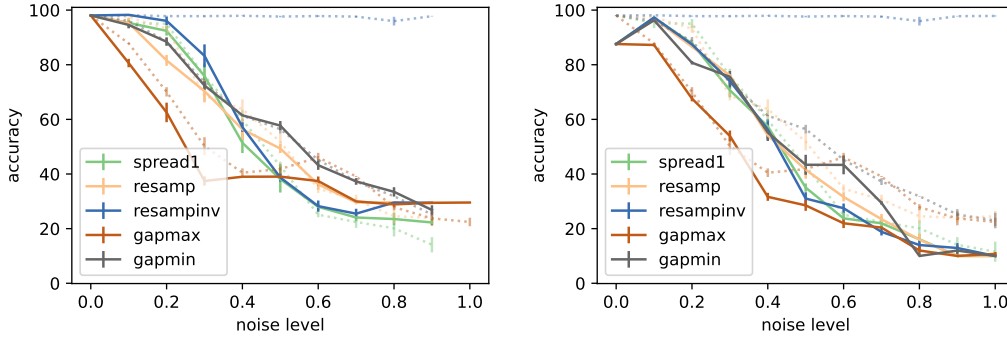

(a) CleanLab (solid) against baseline (dashed)       (b) CoTeaching(solid) against baseline (dashed)

Figure S4: Label noise mitigation methods on synthetic 5D data with 10 classes.

### B.3   Results on Mitigation Methods

Fig. S3 shows the same results as Fig. S3, but with the accuracy results of the vanilla (no label noise mitigation strategy) learning approach in dashed lines to make comparison with this baseline easier.

Fig. S4 compares the clean test accuracy on 10-class, 5-dimensional synthetic data of two label-noise mitigation methods: CleanLab [4] and CoTeaching [1] against the baseline of no mitigation of label noise. We see that broadly, CleanLab has little positive effect, and in particular may perform worse than using no mitigation strategy at low levels of feature-dependent noise; while CoTeaching performs the same or worse than baseline.

## C   Further Image Benchmark Experiments

### C.1   Learning details

These experiments use publicly available code released authors of the various methods on GitHub: CleanLab [4], MixUp [7], Symmetric cross-entropy (SCE) [6], and Co-teaching [1]. Each of the methods is run with default parameters found in the corresponding repositories. The GitHub repositories and the licenses can be found at:

- CleanLab [4] repository us released under MIT license and can be found at
  `github.com/cgnorthcutt/confidentlearning-reproduce`
- MixUp [7] repository us released under MIT and CC-BY-NC licenses and can be found at
  `github.com/facebookresearch/mixup-cifar10`
- SCE [6] repository can be found at
  `github.com/HanxunH/SCELoss-Reproduce`

- Co-teaching [1] repository can be found at
  github.com/bhanML/Co-teaching

Note that SCE [6] and Co-teaching [1] repositories, and the CIFAR10[3] and CIFAR100[3] dataset do not have license attached to them. The CIFAR10[3] and CIFAR100[3] are publicly available on Alex Krizhevsky's website[1], and are a subset of the TinyImages dataset, that is no longer available[2].

Our code is developed on an internal cluster, where each server node is equipped with 4 NVIDIA Tesla A100 cards (each with 40 GB of VRAM), paired with a 64-core AMD EPYC cpu and 256GB of memory. All of our experiments utilize the ResNet-32 architecture across all mitigation methods.

The total amount of compute can be summarized as follows: each experiment with a single mitigation method would on average take 3 hours. Given total of 8 mitigation methods (including 5 variations of CleanLab), 12 noise settings, 2 datasets, and 4 methods of generating noise for each datasets, this results in a total compute requirement of roughly 2304 compute hours.

## C.2  Results on CIFAR

In the following section, we include results for all of the mitigation methods tested: 5 CleanLab variations, MixUp, SCE, and Co-teaching. Each noise mitigation methods was tested on both CIFAR10[3] and CIFAR100[3]. For each dataset, the error flipping was driven by either VisHash or ResNet-50 features, and either by the class noise or feature noise.

Table S1: CIFAR10 clean accuracy with error flipping driven by VisHash features.

| Class noise | $\epsilon = 0.2$ | | | $\epsilon = 0.4$ | | | $\epsilon = 0.6$ | | | $\epsilon = 0.8$ | | |
|---|---|---|---|---|---|---|---|---|---|---|---|---|
| | $s=1$ | $s=2$ | $s=5$ | $s=1$ | $s=2$ | $s=5$ | $s=1$ | $s=2$ | $s=5$ | $s=1$ | $s=2$ | $s=5$ |
| Baseline | 78.15 | 80.01 | 80.80 | 58.94 | 63.49 | 66.88 | 35.26 | 40.62 | 48.70 | **15.07** | **15.28** | 19.44 |
| CL Argmax | 87.86 | 87.57 | 87.44 | **84.66** | 85.48 | 85.62 | 20.81 | **75.08** | 81.95 | 4.78 | 8.46 | **51.48** |
| CL Both | 88.74 | 88.55 | 88.34 | 84.23 | **86.66** | 86.47 | 20.59 | 69.14 | 81.71 | 4.79 | 8.41 | 34.33 |
| CL PBC | 88.20 | 87.93 | 88.17 | 84.62 | 85.89 | 85.79 | 20.59 | 73.99 | **82.22** | 5.09 | 8.07 | 47.41 |
| CL PBNR | 88.33 | 88.53 | 88.41 | 84.65 | 86.45 | 86.58 | 20.78 | 69.00 | 81.91 | 4.91 | 8.33 | 34.41 |
| CleanLab | 88.41 | 88.89 | 88.87 | 83.39 | 86.27 | **86.66** | 21.74 | 73.06 | 81.73 | 4.51 | 7.96 | 50.69 |
| MixUp | **90.62** | **90.45** | **90.69** | 77.77 | 86.33 | 85.01 | 12.27 | 64.29 | 76.02 | 5.16 | 4.59 | 28.16 |
| SCE | 86.21 | 86.62 | 87.38 | 57.42 | 82.71 | 84.23 | 15.53 | 42.86 | 63.47 | 10.68 | 6.26 | 15.90 |
| Co-Teaching | 87.23 | 88.12 | 88.09 | 67.79 | 71.31 | 71.21 | **22.80** | 43.33 | 50.17 | 6.19 | 8.91 | 22.22 |
| **Feature noise** | | | | | | | | | | | | |
| Baseline | 76.92 | 77.43 | 78.73 | 59.86 | 60.84 | 63.72 | 35.89 | 39.49 | 40.70 | 16.81 | 16.87 | 17.55 |
| CL Argmax | 86.96 | 86.75 | 87.09 | **82.50** | 83.71 | 84.69 | 31.44 | 56.33 | 76.64 | **22.86** | **23.29** | **27.71** |
| CL Both | 87.97 | 88.24 | 88.18 | 81.44 | **84.78** | 85.95 | 32.89 | 50.96 | 70.67 | 21.93 | 20.95 | 21.36 |
| CL PBC | 87.95 | 88.02 | 87.96 | 81.55 | 84.11 | 84.65 | 32.59 | 54.12 | **76.74** | 22.18 | 22.63 | 26.12 |
| CL PBNR | 87.75 | 87.76 | 87.75 | 81.30 | 84.67 | 86.01 | 32.06 | 51.66 | 70.43 | 21.88 | 21.58 | 22.06 |
| CL Conf Joint | 88.56 | 88.47 | 88.55 | 82.47 | 84.30 | **86.03** | 32.06 | **56.90** | 76.04 | 22.10 | 22.58 | 26.37 |
| MixUp | **88.70** | **89.85** | **89.83** | 76.14 | 78.30 | 82.48 | **41.75** | 53.59 | 60.15 | 10.96 | 13.08 | 25.43 |
| SCE | 87.26 | 87.65 | 87.39 | 74.71 | 77.70 | 83.68 | 23.97 | 33.65 | 66.98 | 18.81 | 17.03 | 15.52 |
| Co-Teaching | 87.15 | 87.24 | 88.01 | 64.04 | 67.77 | 70.50 | 28.76 | 35.09 | 46.70 | 12.41 | 14.23 | 17.68 |

---

[1] https://www.cs.toronto.edu/~kriz/cifar.html
[2] Please see http://groups.csail.mit.edu/vision/TinyImages/

Table S2: CIFAR10 clean accuracy with error flipping driven by ResNet50 features.

| Class noise | $\epsilon=0.2$ | | | $\epsilon=0.4$ | | | $\epsilon=0.6$ | | | $\epsilon=0.8$ | | |
|---|---|---|---|---|---|---|---|---|---|---|---|---|
| | $s=1$ | $s=2$ | $s=5$ | $s=1$ | $s=2$ | $s=5$ | $s=1$ | $s=2$ | $s=5$ | $s=1$ | $s=2$ | $s=5$ |
| CL Argmax | 87.68 | 87.72 | 87.15 | 84.08 | 85.68 | 85.18 | **28.11** | **76.19** | 81.24 | 4.93 | **16.44** | 41.60 |
| CL Both | 88.51 | 88.36 | 88.42 | 83.45 | 86.19 | 86.44 | 27.08 | 70.65 | 79.86 | 4.47 | 13.76 | 27.20 |
| CL PBC | 88.37 | 88.01 | 88.29 | **84.44** | 86.04 | 85.72 | 26.51 | 74.12 | **81.50** | 4.65 | 15.02 | 40.64 |
| CL PBNR | 88.06 | 88.04 | 88.40 | 83.45 | 86.33 | 86.44 | 27.29 | 70.31 | 79.75 | 4.47 | 14.66 | 26.84 |
| CL Conf Joint | 88.35 | 89.10 | 88.67 | 82.78 | **86.50** | **86.45** | 25.54 | 74.18 | 81.09 | 4.38 | 13.86 | **42.06** |
| MixUp | **90.39** | **90.99** | **90.47** | 77.38 | 84.51 | 84.30 | 11.63 | 64.97 | 66.61 | 4.91 | 5.95 | 30.39 |
| SCE | 86.36 | 87.29 | 87.43 | 50.76 | 82.77 | 84.20 | 12.83 | 35.97 | 61.02 | **10.33** | 5.94 | 11.72 |
| Co-Teaching | 87.07 | 87.61 | 88.29 | 67.15 | 70.70 | 71.70 | 22.21 | 45.73 | 48.29 | 5.50 | 6.42 | 21.21 |
| **Feature noise** | | | | | | | | | | | | |
| CL Argmax | 87.41 | 87.24 | 87.13 | 83.57 | 84.56 | 84.97 | 38.51 | **66.82** | 78.81 | 9.90 | 10.10 | 30.75 |
| CL Both | 88.37 | 88.56 | 88.18 | 82.73 | 85.76 | **86.51** | 42.64 | 59.91 | 73.92 | 9.05 | 10.40 | 20.89 |
| CL PBC | 88.02 | 87.90 | 88.07 | **84.13** | 85.27 | 85.24 | **44.50** | 65.34 | 78.61 | 9.49 | 9.71 | 29.25 |
| CL PBNR | 88.25 | 87.96 | 88.09 | 83.27 | **85.87** | 85.96 | 41.94 | 60.20 | 74.43 | 8.85 | 10.45 | 21.25 |
| CL Conf Joint | 88.63 | 88.89 | 88.69 | 83.03 | 85.72 | 86.29 | 41.58 | 66.40 | **78.86** | 9.01 | 10.07 | **30.88** |
| MixUp | **88.91** | **90.17** | **89.99** | 74.61 | 80.04 | 82.40 | 36.87 | 59.57 | 66.65 | 11.31 | 12.36 | 27.38 |
| SCE | 86.64 | 87.48 | 87.52 | 63.50 | 76.46 | 83.95 | 16.85 | 21.37 | 62.90 | **16.00** | **14.61** | 15.29 |
| Co-Teaching | 86.52 | 87.33 | 88.25 | 65.21 | 66.49 | 69.98 | 27.37 | 34.99 | 47.09 | 9.74 | 9.48 | 16.82 |

Table S3: CIFAR100 clean accuracy with error flipping driven by VisHash features.

| Class noise | $\epsilon=0.2$ | | | $\epsilon=0.4$ | | | $\epsilon=0.6$ | | | $\epsilon=0.8$ | | |
|---|---|---|---|---|---|---|---|---|---|---|---|---|
| | $s=1$ | $s=2$ | $s=5$ | $s=1$ | $s=2$ | $s=5$ | $s=1$ | $s=2$ | $s=5$ | $s=1$ | $s=2$ | $s=5$ |
| Baseline | 57.44 | 57.20 | 57.51 | 42.24 | 45.49 | 47.81 | **24.32** | 29.43 | 34.17 | **10.37** | **10.63** | 15.24 |
| CL Argmax | 51.72 | 50.24 | 50.67 | 33.52 | 42.70 | 44.35 | 9.09 | 18.22 | 31.43 | 3.64 | 4.21 | 7.73 |
| CL Both | 53.39 | 53.92 | 54.40 | 34.28 | 42.89 | 47.55 | 9.55 | 18.53 | 30.21 | 3.56 | 4.37 | 7.31 |
| CL PBC | 52.94 | 52.22 | 52.49 | 34.29 | 43.52 | 45.69 | 9.29 | 18.14 | 32.52 | 3.52 | 4.39 | 7.71 |
| CL PBNR | 53.30 | 53.26 | 53.85 | 35.13 | 42.13 | 47.53 | 8.84 | 18.94 | 29.57 | 3.75 | 4.49 | 6.74 |
| CL Conf Joint | 51.27 | 52.79 | 53.59 | 32.84 | 40.95 | 45.06 | 9.11 | 16.36 | 30.30 | 3.83 | 4.10 | 6.60 |
| MixUp | **64.79** | **65.18** | **65.68** | **50.65** | **57.35** | **58.27** | 18.64 | **34.07** | **48.23** | 5.00 | 7.58 | **17.93** |
| SCE | 54.88 | 56.18 | 54.93 | 33.72 | 46.87 | 50.54 | 12.59 | 19.96 | 36.70 | 6.26 | 8.72 | 12.16 |
| Co-Teaching | 55.85 | 58.66 | 59.65 | 37.79 | 41.83 | 45.29 | 21.11 | 24.43 | 28.00 | 6.72 | 9.28 | 11.57 |
| **Feature noise** | | | | | | | | | | | | |
| Baseline | 53.80 | 54.13 | 54.64 | 37.17 | 38.73 | 39.34 | 20.38 | 21.00 | 22.71 | 7.94 | 8.38 | 8.72 |
| CL Argmax | 48.82 | 49.43 | 49.33 | 32.94 | 37.00 | 39.70 | 10.93 | 15.17 | 21.39 | 6.47 | 6.70 | 7.52 |
| CL Both | 52.79 | 53.83 | 54.41 | 36.27 | 37.42 | 42.58 | 13.30 | 15.25 | 19.86 | 6.49 | 6.72 | 7.20 |
| CL PBC | 50.78 | 51.13 | 52.03 | 34.57 | 38.14 | 41.50 | 11.56 | 15.64 | 22.12 | 6.69 | 6.67 | 7.23 |
| CL PBNR | 52.16 | 51.88 | 53.47 | 34.02 | 37.30 | 41.06 | 12.75 | 15.54 | 19.50 | 6.52 | 6.79 | 7.44 |
| CL Conf Joint | 51.74 | 52.75 | 53.53 | 33.42 | 36.36 | 40.58 | 11.91 | 14.64 | 22.56 | 6.56 | 6.86 | 8.04 |
| MixUp | **63.10** | **62.67** | **63.42** | **47.73** | **50.34** | **51.87** | **27.06** | **29.86** | **33.76** | **9.16** | **10.55** | **13.28** |
| SCE | 53.43 | 54.02 | 55.51 | 32.43 | 36.66 | 42.39 | 9.42 | 11.70 | 17.61 | 7.29 | 7.41 | 7.76 |
| Co-Teaching | 58.40 | 58.16 | 59.28 | 35.71 | 38.93 | 40.80 | 14.12 | 14.40 | 17.05 | 6.17 | 6.28 | 6.08 |

Table S4: CIFAR100 clean accuracy with error flipping driven by ResNet50 features.

| Class noise | $\epsilon=0.2$ | | | $\epsilon=0.4$ | | | $\epsilon=0.6$ | | | $\epsilon=0.8$ | | |
|---|---|---|---|---|---|---|---|---|---|---|---|---|
| | $s=1$ | $s=2$ | $s=5$ | $s=1$ | $s=2$ | $s=5$ | $s=1$ | $s=2$ | $s=5$ | $s=1$ | $s=2$ | $s=5$ |
| CL Argmax | 50.63 | 51.47 | 50.41 | 35.45 | 43.88 | 45.38 | 8.50 | 19.19 | 34.05 | 3.37 | 4.47 | 8.36 |
| CL Both | 53.35 | 54.13 | 54.89 | 35.37 | 44.82 | 48.39 | 9.31 | 19.86 | 31.72 | 3.40 | 5.34 | 8.01 |
| CL PBC | 51.72 | 53.04 | 52.44 | 35.36 | 45.70 | 47.00 | 9.27 | 20.04 | 34.62 | 3.30 | 4.80 | 9.22 |
| CL PBNR | 52.82 | 53.78 | 54.56 | 36.25 | 44.57 | 47.47 | 9.68 | 19.54 | 31.41 | 3.58 | 5.07 | 8.20 |
| CL Conf Joint | 52.07 | 53.71 | 53.94 | 34.32 | 43.17 | 45.61 | 8.76 | 19.22 | 32.15 | 3.27 | 4.49 | 7.59 |
| MixUp | **65.00** | **66.30** | 64.75 | **50.48** | **59.76** | **61.61** | 18.57 | **35.89** | **51.91** | 4.70 | 6.93 | **19.68** |
| SCE | 55.78 | 55.59 | 56.32 | 36.72 | 48.75 | 51.55 | 15.41 | 27.14 | 39.32 | 5.92 | **10.27** | 12.84 |
| Co-Teaching | 56.33 | 58.50 | 59.34 | 38.87 | 42.51 | 44.33 | **20.98** | 25.24 | 29.29 | **6.79** | 9.63 | 12.64 |
| **Feature noise** | | | | | | | | | | | | |
| CL Argmax | 47.14 | 48.46 | 48.65 | 27.19 | 30.85 | 39.33 | 7.92 | 11.61 | 18.08 | 2.55 | 3.26 | 4.05 |
| CL Both | 49.97 | 51.57 | 52.49 | 28.46 | 31.46 | 39.47 | 8.69 | 11.88 | 17.09 | 3.48 | 3.67 | 3.98 |
| CL PBC | 49.34 | 50.87 | 50.67 | 27.96 | 33.26 | 40.22 | 8.42 | 12.76 | 19.33 | 2.64 | 3.14 | 3.95 |
| CL PBNR | 48.93 | 50.93 | 51.75 | 28.67 | 31.51 | 39.49 | 8.46 | 11.83 | 16.54 | 3.17 | 3.72 | 3.72 |
| CL Conf Joint | 48.33 | 49.60 | 51.28 | 26.35 | 30.10 | 38.51 | 8.59 | 11.82 | 18.61 | 2.70 | 3.64 | 4.09 |
| MixUp | **61.20** | **62.00** | **63.56** | **46.47** | **50.63** | **53.01** | **25.01** | **29.42** | **36.18** | **6.62** | **7.03** | **11.67** |
| SCE | 52.31 | 54.30 | 54.41 | 34.20 | 38.41 | 44.19 | 9.16 | 12.75 | 19.25 | 3.82 | 4.57 | 4.81 |
| Co-Teaching | 56.19 | 57.35 | 58.60 | 35.14 | 37.78 | 41.26 | 13.89 | 14.43 | 16.77 | 4.91 | 4.54 | 5.09 |