# OpenReview forum: "Robustness to Label Noise Depends on the Shape of the Noise Distribution"
_NeurIPS.cc/2022/Conference — NeurIPS 2022 Accept_

### Official Review · Reviewer_sFMH · 2022-06-19

**Rating:** 7
**Confidence:** 4
**Soundness:** 4 excellent
**Presentation:** 4 excellent
**Contribution:** 3 good

**Summary:**

The paper suggest difference of generalization ability of Machine Learning considering the scale and shape of the noise distribution. Theoretically and empirically, Machine learning is known to be robust to uniform label noise. However, no study has been researched for the case of class-dependent or instance-dependent label noise setting, and this paper supports theoretical background for those cases. The paper also shows the degradation of noisy label managing algorithms after tipping point empirically, supporting the theoretical result they show.

**Questions:**

1) How did you generate featrure-dependent label noise for CIFAR10 and CIFAR100 for experiments?
2) What will happen if two or more types of noisy label generation algorithm are simultaneously applied? e.g. a noisy labeled dataset with symmetric noise 20% and asymmetric noise 20%.
3) Not to face sudden model performance degradation, is it the only way to get dataset clean enough (not having noisy label as many as the tipping point)?


**Limitations:**

The authors have not addressed ther limitations.

**Strengths And Weaknesses:**

<<Strengths>>
- Originality : The paper theoretically proves the robustness and fragility of ML under trained with noisy labeled dataset setting with different noisy label condition. It is quite interesting. Not only showing the robustness of uniform label noise, the paper also shows the weakness of class- and instance- dependent label noise, which is novel.

- Quality  : The submission is technically sound and the claims are well supported both by theoretical analysis and the expermental results.

- Clarity : Sentences are easy to read. Mathematical expressions are understandable. Well organized.

- Significance : The results are important suggesting two points.
1) The existence of tipping point
2) The model performance degrades suddenly after the tipping point, not gradually.

======================================================================================================

<<Weaknesses>>
- Quality :  Maybe want to see more experimental results?(e.g. Clothing 1M, a well-known benchmark real dataset for noisy label problem, would it also have tipping point?)

- Clarity : (typo) Py|x for section 4.

---

> ### Author Response · Authors · 2022-08-02
> **Author response**
>
> Thanks for clearly summarizing the novel contributions of our paper in the theoretical and empirical analysis of robustness of learning algorithms under various shapes and scale of label noise. We would like to respond to the detailed questions that you pose.
>
> > How did you generate feature-dependent label noise for CIFAR10 and CIFAR100 for experiments?
>
> We use the GapMax function from Table 1. We added this detail and the mathematical formulation of mapping hash distances to estimates of P[Y^*=k | x] to the paragraph describing the data (line 250).
>
> > What will happen if two or more types of noisy label generation algorithm are simultaneously applied? e.g. a noisy labeled dataset with symmetric noise 20% and asymmetric noise 20%.
>
> Interesting question and we can speculate on the answer but we have not tested this. We demonstrate that asymmetric noise is more harmful than symmetric; so in the mixed noise scenario, we expect that the effect of the asymmetric noise will dominate; yet in the specific case suggested (20% + 20%), this is still not enough noise to cross the tipping point.
>
> > Not to face sudden model performance degradation, is it the only way to get dataset clean enough (not having noisy label as many as the tipping point)?
>
> Yes, but cleaning data beyond that point gives little (or no) further improvement.
>
> > Maybe want to see more experimental results?(e.g. Clothing 1M, a well-known benchmark real dataset for noisy label problem, would it also have tipping point?)
>
> This is an interesting suggestion. We focused the paper on more controllable settings in which we know the noise generation process. However as you suggest, with Clothing1M, we can at least vary the ratio of noisy labels within a range because there is a set of 50k clean training labels in that dataset even if we do not know the generating process of the label noise (most real-noise datasets do not provide any clean labels which we need to investigate the tipping point). We can run this experiment and report results in the camera-ready paper, after carefully considering how to structure the experiment because the clean dataset is orders-of-magnitude smaller than the full dataset.
>
> > Clarity : (typo) Py|x for section 4.
>
> Thanks. Fixed.

---

### Official Review · Reviewer_7YEL · 2022-07-11

**Rating:** 3
**Confidence:** 3
**Soundness:** 3 good
**Presentation:** 3 good
**Contribution:** 2 fair

**Summary:**

This manuscript studies the robustness of instance-dependent label noise. The authors provide a theoretical framework that generalizes beyond a typical assumption that label noise is independent of the features by modeling label noise as a distribution over feature space. They also show that the shape of noise distribution has a strong impact on classification performance if the noise is concentrated in feature space.


**Questions:**

In Lemma 3.2, what is the definition of a Bayes optimal classifier, or what is the property of a Bayes optimal classifier? Please formulate it since it is important to understand the following Eq.(6).

Corollary 3.2.1 "The minimum noisy accuracy of $\frac{1}{c}$ occurs at $\epsilon=\frac{c-1}{c}$". This claim is not exactly correct since it misses the condition of uniform assumption. Besides, this claim does no significance since it is common sense that the training labels are useless if $\epsilon=\frac{c-1}{c}$.

Corollary 3.3.1 "Clean accuracy drops by half at $\epsilon=\frac{c-1}{c}$". This claim is not exactly correct. what is the meaning of "Clean accuracy drops by half", is $\epsilon<\frac{c-1}{c}$, and the clean accuracy does not drop by half?

In Fig.3(a), why the greater the $\epsilon$ (noise rate) the better the noisy accuracy?

Eq.(8) is problematic. Since $\sum_{i\neq k}|t_{ki}|^0=s$ and  $\sum_{i\neq k}t_{ki}=1$, $\sum_{i=1}^{k} \eta_{ki} \neq 1$ if $s\geq 2$.

From the process of generating feature-dependent noise (line186-188), it seems a class-dependent label noise rather than feature-dependent noise.


**Ethics Review Area:**

["I don’t know"]

**Limitations:**

Please see the comment above.

**Strengths And Weaknesses:**

The theories (Lemma3.2 and Lemma3.5) presented in this paper simply derived the results from [2] under both uniform label noise and class-dependent label noise, while theoretical analysis on instance-dependent label noise is not provided, the novelty is limited.

This paper focus on the instance-dependent label noise studied, but two-thirds of the paragraph are describing instance-independent label noise.

[2] Understanding and Utilizing Deep Neural Networks Trained with Noisy Labels.

---

> ### Author Response · Authors · 2022-08-02
> **Author response**
>
> The review summary points out that we “provide a theoretical framework that generalizes beyond a typical assumption that label noise is independent of the features” and that we “show that the shape of the noise distribution has a strong impact on classification performance”.
>
> We find the questions about the contribution of the paper to be enlightening and so we have added more discussion to the paper to clarify why the generalizations that we provide beyond previous work are important for the future study of label noise. The detailed response below starts with the questions about the contribution of the paper.
>
> While the Soundness rating is “3 good” the written comments initially led us to believe that Reviewer 7YEL found problems in our theory. After re-reading the review, it is clear that the word “incorrect” is used to describe issues of “clarity”. We do not see anything in this review that disputes the correctness of the paper’s claims despite the use of the word “incorrect”. We have addressed the clarity issues as detailed below.
>
> > The theories (Lemma3.2 and Lemma3.5) presented in this paper simply derived the results from [2] under both uniform label noise and class-dependent label noise, while theoretical analysis on instance-dependent label noise is not provided, the novelty is limited.
>
> Our results are more general than in [2], as noted originally in line 113. To clarify things and reinforce the impact of the generalization, we have added a more in-depth discussion to section 3.5 lines 178-183.  [2] provides a simpler claim assuming that the model has "sufficiently high capacity", which ignores the effect of the baseline model errors (where by "baseline model" we mean if it were trained on clean data, $m^*_k(x)$). This is an important distinction because the ability to correctly predict a label (or whether a label is noisy) does depend on the baseline performance of the model not just the level of noise (even without noise, the model is sometimes wrong). This makes the dependence on x explicit: even in the case of uniform and class noise, model errors depend on x and so the ability to be robust to noise depends on x.
>
> > In Fig.3(a), why the greater the (noise rate) the better the noisy accuracy?
>
> Lines 219 - 221 discuss this phenomenon of the model fitting the noise (which increases noisy accuracy) rather than the clean data as the amount of noise increases: "Beyond the tipping point (the minimizing point in the quadratic function of the noisy accuracy), the neural network begins fitting the noise instead of the true signal...". We added a similar discussion to Section 3 line 115 (much earlier in the paper where Figure 3 appears) because your question makes it clear that this may not be obvious to readers when first seeing Figure 3.
>
> > Corollary 3.2.1 is incorrect because it is missing assumption of uniform noise; and it has no significance because it is common sense.
>
> We added the assumption explicitly to Corollary 3.2.1 as suggested, even though it already appears in the line above as part of Lemma 3.2. We disagree with the statement that it is common sense. It may be intuitive that this is the point at which the level of label noise overwhelms a classifier; but we provide the mathematical derivation based on minimizing the quadratic function given in Eq. 6. Especially given the question about noisy accuracy increasing in Figure 3a (which is a plot of Eq 6); it does not seem that it was obvious that there would be a minimizable function from which to analytically derive Corollary 3.2.1.
>
> > Corollary 3.3.1 is not exactly correct.
>
> This is a little difficult to respond to without more information. We have checked Corollary 3.3.1 and it is correct. We have clarified it in the paper by making the uniform noise assumption explicit (similar to Corollary 3.2.1); and we changed the phrasing "accuracy drops by half" to be more explicit with "accuracy drops to $\bar{m}^*_k / 2$". Does this clarify the Corollary or is there still an issue to address?
>
> > Eq(8) is problematic.
>
> Reviewer STc1 also pointed out this typo. The $t_{ki}$ values should not be divided by s. We fixed this typo in Eq 8; verified the derivation of Eq 9 is correct; and validated that our code performs the correct normalization for our experiments.
>
> > From the process of generating feature-dependent noise (line186-188), it seems a class-dependent label noise rather than feature-dependent noise.
>
> Our feature-dependent noise is dependent on both features and class; which is the most general dependency relationship. The standard terminology of "class-dependent label noise" implies that the label noise is conditionally independent of the features given the class and we clearly define this in both the Introduction and the Problem Statement. We call our noise "feature-dependent label noise" to distinguish it from the standard assumption of conditional feature independence.

---

> > ### Comment · Reviewer_7YEL · 2022-08-05
> > **There remains some questions should be addressed**
> >
> >
> > The title ``Robustness to Label Noise Depends on the Shape of the Noise Distribution in Feature Space'' shows that this manuscript studies the Shape of the Noise Distribution in Feature Space. While uniform label noise and class-dependent label noise are label noise in label space instead of feature space. Is this title misleading?
> >
> > How to deal with label noise based on those theoretical results is still unclear. The contribution of this paper is marginal.
> >
> > Lack of feature-dependent label noise theoretical analysis.
> >
> > The process of generating feature-dependent label noise is still unclear, more details show be provided for reproducing experimental results, not only an introduction as shown in Table 1.
> >
> > Line 117, Eq 6 should be Eq. (6) for more clarification.
> >
> > Line 114, ``a Bayes optimal classifier'' in Lemma 3.2, is still unclear what properties with it.
> >
> > Line 125, Corollary 3.1, supposing \bar{m}_{k}^{\ast}=1 (as describe in the first submission, line 113), according to  Corollary 3.1, a model trained with noisy labels at noise level \epsilon=\frac{c-1}{c} can achieve a clean accuracy of \bar{m}_{k}^{\ast}/2=1/2 ? I don't agree with this claim since a model can only random guess a label for each instance when noise level at \epsilon=\frac{c-1}{c} under the uniform assumption.
> >
> > ​Are those experimental results trained under a large enough learning rate? Label noise is sensitive to the learning rate. Such an important hyperparameter should be provided.

---

> > > ### Author Response · Authors · 2022-08-09
> > > **Author response**
> > >
> > > We appreciate the opportunity to further discuss the revised paper base on your detailed comments.
> > >
> > > > Title?
> > >
> > > To address this concern, we can modify the title to “Robustness to Label Noise Depends on the Shape of the Noise Distribution” (removing “in Feature Space”). We do show novel results on the effect of feature-dependent noise in both synthetic and benchmark data while considering different shapes of the feature-dependent noise distribution. We also present results on uniform and class-dependent noise, so the more general title does sound more appropriate.
> > >
> > > > How to deal with label noise based on those theoretical results is still unclear.
> > >
> > > In fact our theoretical results indicate that “dealing with label noise” is often unnecessary, especially if the ML research community continues to assume feature-independent noise.
> > >
> > > > a model trained with noisy labels at noise level \epsilon=\frac{c-1}{c} can achieve a clean accuracy of \bar{m}{k}^{\ast}/2=1/2 ?
> > >
> > > No. $\bar{m}{_k}^{\ast}/2$ would only be $1/2$ if $\bar{m}{_k}^{\ast} = 1$. Furthermore, it decreases steeply beyond this point; as plotted in Figures 3b, 4b, and 4d. This is just the midpoint of the value of the logistic function. The minimum in clean accuracy is worse than random-chance because the label noise is defined (as is typical in existing literature) such that the true labels become less (not equally) likely than the wrong labels at high levels of label noise.
> > >
> > > > learning rate?
> > >
> > > Supplement lines 75-77 for the synthetic data describe using cross-entropy loss with the Adam optimizer with learning rate 0.001. We did need to cross-validate various loss functions, optimizers and learning rates to find a combination that works well (as mentioned in line 77). Supplement line 105 describes the settings for the benchmark data, in which we use the learning parameters chosen by the authors of the methods we compare.
> > >
> > > > Line 117, Eq 6 should be Eq. (6) for more clarification.
> > >
> > > Thanks for noting typos! We fixed this and similar missing parentheses.
> > >
> > > > Line 114, ``a Bayes optimal classifier'' in Lemma 3.2, is still unclear what properties with it.
> > >
> > > We will add this clarification, taking care to match notation already laid out.

---

### Official Review · Reviewer_xHwG · 2022-07-12

**Rating:** 7
**Confidence:** 3
**Soundness:** 3 good
**Presentation:** 3 good
**Contribution:** 3 good

**Summary:**

The authors analyze the problem of the robustness of the machine learning classifiers to the label noise under different noise types: 1) uniform (independent of features and classes), 2) class-dependent, and 3) feature-dependent. In their analysis, the authors focus on analyzing the "typing point" - the noise ratio value below which classifiers suffer a significant reduction in predictive performance. The authors show that typing point can be much lower when noise is feature dependent than in two other cases, suggesting that one needs to be careful when assuming that the noise is independent of features. The theoretical results are later confirmed by empirical evaluation on both synthetic and real datasets with introduced noise.

**Questions:**

I don't have questions right now.

**Limitations:**

Since this isn't a methodology paper, there are not many limitations to discuss. Also, there are no potential negative social impacts of this work.

**Strengths And Weaknesses:**

Strengths:
- Although this isn't a methodology paper, and while the main claim is not surprising, the paper helps to understand and provide a sound analysis of the problem. This is the type of paper that is interesting to read as opposed to many papers that demonstrate yet another epsilon improvement in an experimental evaluation.
- The theoretical results are confirmed by empirical experiments, on different datasets and with different types of introduced noise.
- The experiment with methods for learning with noise labels is a nice addition.

Weaknesses:
- Most of the presented results come from previous works.
- The main conclusions are not surprising and, as the authors acknowledge, observed in previous empirical studies.

NITs:
- In eq 14. I think double equal signs are not needed.
- In line 180, P Y|X are missing brackets.

---

> ### Author Response · Authors · 2022-08-02
> **Author Response**
>
> Thank you for the detailed review. You did not ask any specific questions, but we identified a few of your sentences that we would like to comment on.
>
> > The paper helps to understand and provide a sound analysis of the problem… The theoretical results are confirmed by empirical experiments, on different datasets and with different types of introduced noise.
>
> Thanks. Yes, deeper understanding of the label noise problem is the primary goal of the paper.
>
> > The experiment with methods for learning with noise labels is a nice addition.
>
> Because you and Reviewer sFMH found this empirical approach interesting, we added more specifics about our method for generating feature-dependent noisy labels which may be useful to future empirical studies.
>
> > Most of the presented results come from previous works.
>
> Which previous works have the same results? Our results are more general (as noted in line 113) than our reference [2]. We discuss the differences between our theoretical results and other works specifically in Section 3.5; as well as the broader related work in Section 2.2.
>
> > The main conclusions are not surprising and, as the authors acknowledge, observed in previous empirical studies.
>
> Sort of. We explicitly point out where our theory matches known empirical results and where it helps to explain seemingly diverging results: some studies show how severely damaging label noise is while other studies show robustness even to high levels of noise - but we show that they are looking at different points on a curve that has a tipping-point which has not been made explicit previously. Furthermore, we have contributed a novel approach to generate feature-dependent noise for an empirical study; where we have found only 2 existing empirical studies with feature-dependent noise [8,10], but as noted in our Related Work section, these approaches change the features themselves when introducing the noise and so there is a domain shift in addition to the label noise.
>
> > In eq 14. I think double equal signs are not needed. In line 180, P Y|X are missing brackets.
>
> Thanks, fixed.

---

> > ### Comment · Reviewer_xHwG · 2022-08-10
> > **Thank you for your response**
> >
> > Dear Authors,
> >
> > Thank you for your comments, even though I haven't asked any questions. I also appreciate your responses to other reviewers. I haven't been able to go through the revised paper yet, but I plan to do it soon. Regarding my vague (sorry for that) comment: "Most of the presented results come from previous works." Yes, I meant [2]. I agree with the view of Reviewer 7YEL, that most of the theory you present is actually related to instance-independent label noise (and I agree with Reviewer 7YEL's claim about the title) and that these results are derived from [2]. But I like presented generalization and don't see it as a big issue.
> >
> > Kind regards

---

### Official Review · Reviewer_sTc1 · 2022-07-12

**Rating:** 7
**Confidence:** 3
**Soundness:** 3 good
**Presentation:** 3 good
**Contribution:** 2 fair

**Summary:**

This paper studies how different forms of label noise affect the performance of ML classifiers. It is argued that the most benign form of label noise is uniform noise independent of features and labels, and the most harmful is feature dependent label noise. For feature independent label noise, classifiers retain good accuracy until some level of noise, after which there is a sharp drop-off. On the other hand, for feature dependent label noise, what matters is whether the flips occur close to or far away fromt he decision boundary. This is illustrated through theory and experiments.

**Questions:**

I have asked a few questions in the strengths and weaknesses section, as well as the limitations section.

The \eta_{ki} values in eq. 8 do not seem to add up to 1? Could you please clarify.

I like the paper and would be happy to increase my score from 6 to 7 if the response is strong.


**Limitations:**

I feel that this paper does not address the 'so-what' question: how do I use this theory in a practical setting where I don't know the extent of label noise, or the type of label noise? I encourage the authors to think and elaborate on this.

**Strengths And Weaknesses:**

For the theory, the authors study how the Bayes optimal classifiers with and without label noise compare for simple label flipping attacks. While studying the Bayes optimal classifiers is quite limiting, it is nonetheless a useful starting point that seems to not have been considered before in the literature. The theory is simple, but leads to the following insights:
- Feature-independent label flipping is benign until a large value of epsilon, after which there is a sudden drop in performance. In particular, for class-dependent label flipping, the point of drop in performance depends on the 'spread' of the flip (the support size of the flip targets).
- Feature dependent label flipping can hurt even at a small value of epsilon, especially if the flipped labels are closed to the optimal decision boundary. On the other hand, flips away from the decision boundary have a smaller effect.

Despite the simplicity of the theoretical analysis, it matches very well with synthetic experiments.

The results on CIFAR-10 and CIFAR-100 (Tables 2 and 3) seemed more of a mixed bag. Shouldn't the drop in performance from class noise to feature noise be more based on the theory? Why is this not the case?

The paper was easy to read and follow, although I felt that the authors could have been more transparent about the limited scope of the theoretical study (only Bayes optimal classifiers).

Overall, I found the paper informative, but lacking in a final punchline message. The existing findings are concrete and well-supported, but not too surprising (at least to me): I would have expected that feature based label noise is a more powerful attack than class based or uniform noise. The work makes no methodological contribution, which for me slightly increases the bar of acceptance. (Perhaps this research could help engineers seeking annotations to take appropriate steps if label noise seems higher than a threshold? Not sure.)

---

> ### Author Response · Authors · 2022-08-02
> **Author response**
>
> We appreciate the excellent summary of the theoretical insights drawn from this paper and the note that our empirical results validate the theoretical results. The major question is about how to put this theory into practice and we address that in our detailed responses below.
>
> > How do I use this theory in a practical setting where I don't know the extent of label noise, or the type of label noise?
>
> The theory is most helpful in designing evaluation approaches for label noise detection and mitigation. We now know what levels of noise are most helpful to evaluate (near where the tipping-point occurs), that class-dependent noise is quite similar to uniform noise spread across just a few classes (so there is no need to evaluate various levels of spread of class-dependent noise), and that incorporating feature-dependent noise is most informative for practical applications. Focusing evaluation on the most informative experiments will significantly reduce the computational burden of empirical studies, especially when we consider that label noise mitigation strategies can require training 2 models (such as for CoTeaching) or even 6 models (such as for CleanLab).
>
> The theory is also helpful in explaining why learning from massive, noisy datasets can be so successful, especially when the noisy dataset is used as a pre-training task: major patterns in data can be learned despite label noise as long as there is some signal within the noise. Our theory can also be used to guide the curation of datasets as they are being collected. Not all labels need to be clean, but enough labels in every class across the relevant feature space do need to be clean. Data-specific approaches may look for regions of feature space where labels tend to be noisy and focus more effort on clean labels for those regions.
>
> This discussion is now included in Section 6 with a mention in the Introduction.
>
> > The results on CIFAR-10 and CIFAR-100 (Tables 2 and 3) seemed more of a mixed bag. Shouldn't the drop in performance from class noise to feature noise be more based on the theory? Why is this not the case?
>
> Tables 2 and 3 for the most part do reflect our findings from synthetic data and theory. It is interesting to note that when a mitigation strategy is performing poorly (e.g. SCE on CIFAR10 at $\epsilon=0.2$ or CleanLab on CIFAR100 at $\epsilon=0.2$) compared with the other methods; then it does not seem to matter whether the noise is feature-dependent or class-dependent. It's true that on CIFAR10, the accuracy drop from class noise to feature noise at $\epsilon=0.2$ is small or non-existent; where we would expect this to be larger. However, on CIFAR100 there is a performance drop at $\epsilon=0.2$ and for both datasets the performance drop is even larger for $\epsilon=0.4$; and for $\epsilon=0.6$ while $s=\{2,5\}$; and for $\epsilon=0.8$ while $s=5$; which is consistent with our theory. As we show in Figure 7 on the synthetic data, the noise label mitigation strategies CleanLab and CoTeaching do reduce the gap in performance between class noise and feature noise. We have added this discussion to the Results section (lines 253-264).
>
> > The $\eta_{ki}$ values in eq. 8 do not seem to add up to 1?
>
> Good catch! The $t_{ki}$ values should not be divided by s. We fixed this typo in Eq 8; verified the derivation of Eq 9 is correct; and validated that our code performs the correct normalization for our experiments.
>
> > Authors could have been more transparent about the limited scope of the theoretical study (only Bayes optimal classifiers).
>
> Thanks for pointing this out. This limitation is added into the Conclusion.

---

> > ### Comment · Reviewer_sTc1 · 2022-08-04
> > **Increased score from 6 to 7**
> >
> > I haven't been able to go through the updated paper in detail but it appears that the authors have made a concerted effort in addressing the 'so-what' question I had. I have increased my score from 6 to 7 in good faith, as promised.

---

### Meta-Review · Area_Chair_82hu · 2022-08-26

**Recommendation:** Accept
**Confidence:** Certain

**Metareview:**

The authors provide an interesting perspective on robustness by looking at how different forms of label noise affect performance. Although some of the results are unsurprising given the literature, providing theoretical justification for empirically observed phenomena is an important contribution that could promote new ideas. The paper is also well-written and was appreciated by most reviewers.

**Award:**

No

---

### Decision · Program_Chairs · 2022-09-14

Accept